# Identification and Detection of Bioactive Peptides in Milk and Dairy Products: Remarks about Agro-Foods

**DOI:** 10.3390/molecules25153328

**Published:** 2020-07-22

**Authors:** Himani Punia, Jayanti Tokas, Anurag Malik, Sonali Sangwan, Satpal Baloda, Nirmal Singh, Satpal Singh, Axay Bhuker, Pradeep Singh, Shikha Yashveer, Subodh Agarwal, Virender S. Mor

**Affiliations:** 1Department of Biochemistry, College of Basic Sciences & Humanities, CCS Haryana Agricultural University, Hisar 125 004, Haryana, India; himanipunia91@gmail.com; 2Department of Seed Science & Technology, College of Agriculture, CCS Haryana Agricultural University, Hisar 125 004, Haryana, India; anuragmalikseed@hau.ac.in (A.M.); nirmal.icar@gmail.com (N.S.); bhuker.axay@gmail.com (A.B.); pradeepkhilery29@gmail.com (P.S.); virendermor@gmail.com (V.S.M.); 3Department of Molecular Biology, Biotechnology and Bioinformatics, College of Basic Sciences & Humanities, CCS Haryana Agricultural University, Hisar 125 004, Haryana, India; sonalisangwan03@gmail.com (S.S.); shikhayashveer@gmail.com (S.Y.); 4Department of Horticulture, College of Agriculture, CCS Haryana Agricultural University, Hisar 125 004, Haryana, India; s_baloda@rediffmail.com; 5Forage Section, Department of Genetics & Plant Breeding, College of Agriculture, CCS Haryana Agricultural University, Hisar 125 004, Haryana, India; satpal.fpj@gmail.com; 6Department of Business Management, College of Agriculture, CCS Haryana Agricultural University, Hisar 125 004, Haryana, India; subodh.agarwal47@gmail.com

**Keywords:** bioactive peptides, milk, dairy products, milk proteins, functional foods

## Abstract

Food-based components represent major sources of functional bioactive compounds. Milk is a rich source of multiple bioactive peptides that not only help to fulfill consumers ‘nutritional requirements but also play a significant role in preventing several health disorders. Understanding the chemical composition of milk and its products is critical for producing consistent and high-quality dairy products and functional dairy ingredients. Over the last two decades, peptides have gained significant attention by scientific evidence for its beneficial health impacts besides their established nutrient value. Increasing awareness of essential milk proteins has facilitated the development of novel milk protein products that are progressively required for nutritional benefits. The need to better understand the beneficial effects of milk-protein derived peptides has, therefore, led to the development of analytical approaches for the isolation, separation and identification of bioactive peptides in complex dairy products. Continuous emphasis is on the biological function and nutritional characteristics of milk constituents using several powerful techniques, namely omics, model cell lines, gut microbiome analysis and imaging techniques. This review briefly describes the state-of-the-art approach of peptidomics and lipidomics profiling approaches for the identification and detection of milk-derived bioactive peptides while taking into account recent progress in their analysis and emphasizing the difficulty of analysis of these functional and endogenous peptides.

## 1. Introduction

Milk is a vital food that satisfies the neonate’s nutritional needs and its composition is formulated to facilitate the survival of the species as a result of 200 million years of evolution [1]. Milk is a source of lactose, lipids and proteins, as well as vitamins, mineral substances, oligosaccharides, intrinsic immune factors, immune globulins, hormones, enzymes, and neonatal growing factors [2]. Such components play a significant role in different body functions like cardiovascular, immunomodulation, metabolic and neuronal growth and in the establishment and implementation of a gut microbiome [3,4,5,6]. The milk industry is the most diverse and intensively developed global dairy sector leading to the diversification of targeted dairy formulations aimed at different age groups, in order to meet particular nutritional needs. A report of the United States Department of Agriculture (USDA) Global Agricultural Information Network estimated a 38 million ton increase in liquid milk use, marking a significant increase concerning a historically low milking base, given that customers are acquiring a new taste for milk. Milk and milk products are vital to human nutrition and are known as carriers of amino acids, proteins, fat, water-soluble vitamins, calcium, essential fatty acids, and several other bioactive compounds of significant importance in diverse biochemical and physiological functions [7].

The primary sources of bioactive peptides are milk proteins. These bioactive peptides are short amino acid sequences encrypted in milk proteins and may be released by *in vivo* fermentation of milk with a proteolytic starter (lactic acid bacteria) or by enzyme hydrolysis during gastrointestinal digestion [8,9].They remain inactive within the primary structure of milk proteins and are released by casein proteolysis [10]. The bioactive peptides released are often small and range in size from 2 to 50 amino acid residues [11]. A number of these peptides found in bovine milk or synthesized *de novo* have been shown to affect the cardiovascular, neurological, digestive, endocrine and immune systems while displaying functional properties like anti-microbial, antithrombotic, anti-hypertensive, anti-atherogenic, anti-oxidant and immunomodulatory activities [6,12,13,14]. Since hypertension is becoming more prevalent globally, researchers have focused mainly on the evaluation of antihypertensive peptides and their intervention in cardiovascular diseases [8]. In therapeutic implementations, the potential effect of these anti-hypertensive peptides has been well documented [15,16]. Moreover, anti-hypertensive activity is one of most important functionalities of peptides but additional applications of dairy peptides can also be pinpointed, despite being less researched. Simultaneously, the increasing consumer interest in solving health issues by making conscious eating choices has led to food containing naturally occurring peptides rather than synthetically produced pharmaceuticals being preferred, especially if comparable effectiveness for targeted applications has been proven.

Analytical approaches for milk component analysis are moderately uncommon in the field of probabilistic analytical chemistry. Dietary peptide identification has been investigated mainly by subjecting food to *in vitro* digestion simulation [17,18,19]. Table 1 summarizes the different analytical techniques used for identification and detection of milk and dairy components actively involved in human health. Over past 25 years, statistical tools for characterization and the validation of analytical method efficiency in combination with the enhancements in both instrumental and chemical approaches have achieved better testing performance. Lynch and coworkers [20] reported that methods for calculating the composition of milk in quantitative analytical chemistry are relatively uncommon since the findings determine the split of substantial sums of money between milk sellers and buyers. There is, therefore, an excellent opportunity to improve and optimize these techniques in order to achieve a degree of research rigor rarely applied to other approaches or by laboratory workers employed in analytical chemistry. Over the past 25 years, mid-infrared (MIR) milk analysis and improved performance of chemical processes for the calibration of MIR has helped the milk and dairy industry to change from weight and fat-dependent milk payment to weight-based milk assessment (e.g., fat, actual proteins, other solids) for every milk component delivered. These advances have been made possible by research in order to enhance both the chemical and instrumental analytical approaches for milk components.

Nevertheless, the initiative, leadership and financing for most of the developments in the testing of milk in the USA during the last 25 years came from the USDA Test Procedures Committee, Dairy Programs, and the Federal Milk Market orders.

The consolidation and increased production of milk products have led to the introduction of new, fast, reliable and cost-effective methods for quality assurance and composition analysis. The sensory and instrumental evaluation of dairy food characteristics could be used for quality assurance determinations prior to the transition of the product to the customers, who perform the final sensory assessment, although it is always important to detect, identify and correct product defects. Modern instrumental and sensory strategies in dairy food production facilitate the optimization of a product’s desired attributes and the creation of new products that suit the product’s future market segment attribute profile. The sensory evaluation of dairy products has developed from both defect-oriented recognition system and intensity-development approaches that explain several more dimensions of the characteristics of a product. The data from sensory processes like quantitative descriptive analysis (QDA) can be used for defining and optimizing the characteristics of a dairy food product by combining them with the aim of measurements from a chemical (e.g., gas chromatography) or physical (e.g., rheology measurements, such as dynamic mechanical analyses) method. For the analytical and instrumental data processing, more sophisticated statistical methods such as principal component analysis are frequently used for the data analysis. Comprehending the underlying presence of milk and dairy bioactive peptides, may, therefore, reveal possibilities of custom-made processing to enhance identification of milk peptides and produce enriched milk products that improve health aspects. This review focuses on the methodologies used for the analysis of milk, and dairy products will enhance our understanding of dairy products’ bioactive properties and direct the future identification of new functional products. This article also presents state-of-the-art approaches for complementing and profiling of the bioactive detection, regardless of the small proportion of amino acid units.

## 2. Primary Classes of Milk-Derived Bioactive Peptides

Dairy products comprise about 5–10% of individuals’ average diet and may vary indifferent geographic regions, i.e., 3% of dietary energy in Asia and Africa; 8–9% in Europe [44,45,46]. Dairy and milk products are nutritious foodstuffs containing various essential nutrients such as vitamins, oleic acid, linoleic acid, omega-3 fatty acids, minerals, and bioactive agents such as antioxidants [47]. The consumption of antioxidants in the form of foods that are high in antioxidants can prevent oxidative stress and damage to the body [48]. Milk and milk products possess antioxidant potential which might be due to the presence of sulfur-containing amino acids, for example, methionine and cysteine, vitamins A and E, carotenoids, and antioxidative enzymes (superoxide dismutase, catalase and glutathione peroxidase), and appreciable amounts of a polyphenolic metabolite of daidzen [49]. Milk includes components which provide neonates and adults with essential nutrients, immunological defenses and biologically active substances. In general, α-LA, β-LG, immunoglobulins, protease-peptide fractions, caseins, lactoferrin, and small fractions of whey proteins such as serum albumin and transferrin are the main fractions of the protein inside bovine milk. Bioactive peptides can be produced *in-vivo* via gastrointestinal processes. Sometimes, because of their hormone-like properties, their liberation affects various physiological reactions. These peptides can also be formed in-vitro by enzyme hydrolysis, encoded into native protein precursor sequences. The peptides are purified by different separation techniques from protein hydrolysates and their bioactivities are then assessed. There is substantial proof that several bioactive peptides perform a multifunctional role, often sharing similar structural characteristics centered on a biospecific, specified role. Bioactive peptides have been identified within the amino acid sequences of milk proteins, Hydrolytic reactions, including digestive enzyme-catalyzed reactions, lead to their release. These peptides directly influence several biodynamic processes that evoke gastrointestinal, immunological, behavioral, nutritional and hormonal reactions. There are well-characterized bioreactions connected with any of the metabolic groups. Figure 1 depicts the different classes of milk-derived bioactive peptides and their physiological roles in human health. Numerous applications of bioactive peptides in the preparation of functional foods have already been developed. For example, phosphopeptides extracted from casein fractions are commonly used as both dietary and medicinal supplements. Adding bioactive peptides to food products may potentially improve consumer health due to their antimicrobial properties. Finally, bioactive peptides may function as health products and provide therapeutic value for either infection control or disease prevention.

### 2.1. Antimicrobial Peptides

The overall antibacterial activity in milk is higher than would be expected from the amount of the various immunoglobulin and non-immunoglobulin protein contributions. This is most probably due, at least in part, to their synergy. In addition to the inactive protein precursors, a further contributing factor may be the presence of naturally occurring bactericidal peptides. The early literature identified lactoferrin as an antimicrobial milk protein. At that time, studies also detailed the discovery of the essential glycopeptides with bactericidal activity against different strains of *Staphylococcus aureus* and *Streptococci*. Their interest as a commercial antimicrobial drug has in general been overlooked. In recent times, however, renewed interest has been shown in exploiting bioactive peptides in the healthcare industry. They have inhibitory action against various Gram-positive and Gram-negative bacterial strains viz. *E. coli*, *Aeromonas hydrophila*, *S. typhi*, *S. enteritidis*, *Bacillus cereus*, *Staphylococcus aureus*, etc. [50]. Similarly, digestion of protein with chymosin produces caseicidin peptides that displayed antimicrobial activities against several bacterial strains such as *Streptococcus pyogenes*, *Staphylococcus* spp. and *B. subtilis* [51].

Peptides with antimicrobial activities are listed in Table 1. Casocidin-I, a cationic fragment released from casein digestion can inhibit the growth of *S. carnosus* and *E. coli* [52] while two other bioactive peptides fromcasein, viz.f164–179 and f183–207, inhibit several different bacterial strains [53]. Similarly, digestion of lactoferrin releases a lactoferrampin fragment that exhibitsbactericidal activity against *E. coli*, *Pseudomonas aeruginosa*, *B. subtilis* and *Streptococcus mutans* [54]. Studies on chymosin bioactive peptides have identified new antibacterial peptides, namely, isracidine (αs_2_-CN bovine) against *S. pyogenes*, *Listeria monocytogenes* and *S. aureus* [51]. The specific cleavage of casein releases glycomacropeptides from chymosin [55] which exhibit antimicrobial activity against *E. coli* and *S. mutans* while glycomacropeptide activates the intestinal microflora [56]. Lactoferricin B and Isfracidin exhibited inhibitory action against *Candida albicans* [51,57]. At the same time, lactoferrin and its derivatives display inhibitory activity under *in vitro* conditions against several pathogenic bacterialstrains, such as *Salmonella enteritidis*, *Haemophilus influenzae*, *Clostridium perfringens*, *P. aeruginosa*, L. *monocytogenes*, *S. aureus*, *C. albicans*, *Vibrio cholerae*, *Helicobacter pylori*, and also antiviral activities against poliovirus, hepatitis C, B and G, herpes simplex virus, HIV-1, and rotavirus [58,59].

### 2.2. Angiotensin-Converting Enzyme (ACE)/Anti-Hypertensive Inhibitory Peptides

Angiotensin-converting enzyme (ACE) is a peptidyl dipeptidase enzyme with a capability to cleave the carboxyl-end of substrate which upon conversion from angiotensin I into angiotensin II (an active peptide hormone), can control the rise in blood pressure. This stimulates aldosterone release, leading to higher levels of sodium and increased blood pressure. However, ACE may suppress blood pressure development in the form of an antihypertensive peptide [60]. Milk proteins have been characterized as a range of inhibitory ACE peptides and effective ACE inhibitors of milk casein (casokinins) and whey proteins (lactokinins) [61,62,63]. Two effective inhibitory tripeptides: Ile-Pro-Pro (IPP) and Val-Pro-Pro (VPP) derived from bovine casein had been isolated from sour milk fermented with *Saccharomyces cerevisiae* and *L. helveticus* [64]. Casein peptide, VPP, has been identified as a potential ACE inhibitor. Studies indicate that monocyte adherence to inflamed endothelia has been well moderated by VPP that might prove essential in atherosclerosis prevention [65]. Several *in vivo* studies conducted on hypertensive rats and humans have shown that certain ACE inhibitory peptides have lowered the blood pressure in a dose-dependent manner after oral or intravenous use [66,67]. ACE inhibitors at the end of their C-terminal are the di-peptides or tri-peptides containing proline, arginine, and lysine amino acids. The bioactive sequence of amino acids with anti-hypertensive activity has been isolated from human and bovine casein protein. Whey proteins isolated from lactic acid bacteria such as *L. lactis* and *L. helveticus* were immune to endopeptidases of the digestive tract, as a result, can be quickly absorbed from the blood circulation [68]. The inhibitory peptides of ACE, such as j-casein, and β-casein, were isolated from the enzymatic digestion of acidic proteins αs_1_ and β-CN [69]. ACE inhibitory peptides, like, α-lactorphine and β-lactorphine are released from whey proteins, such as, lactoglobulin and α-lactalbumin as well as casein derived peptides [70,71,72]. The peptides EMPFPK and YPPEPYTE were derived from sequences of casein f(108–113) and casein f(114–121), which exhibited *in vitro* inhibitory effects for ACE [73]. ACE peptides inhibitors are natural foods preventative used to regulate elevated blood pressure, which can lead to a reduction in the need for medicines that have strong side effects.

### 2.3. Immunomodulatory Peptides

Glycopeptides, hormones, and peptides fragments derived form immunoglobulins (Ig) are generally defined as the immunomodulatory bioactive peptides, which mediate the humoral and cell-mediated body’s immune system. Subsequently, numerous bovine β-casein peptides were identified, causing phagocytic reactions in human beings and inhibiting *K. pneumoniae* transmission in mice under *in vivo* conditions [74]. Recently, several cytochemical studies have suggested that both casein and whey peptides have been produced for immunomodulatory peptides. The activation and proliferation of proteins were associated with macrophage phagocytic involvement, antibody formation and human lymphocyte synthesis and cytokine regulation [75,76]. Peptides made from cytomodulatory products of casein can inhibit cancer cell growth through activation and stimulation of competent immune system cells [77]. The immuno-suppressive impacts on IgG antibody synthesis and their derivatives have shown to be important immune co-modulatory functions of glyco-macropeptides [56,78]. Digestion of lactoferrin releases lactoferricin B, which allows interaction with neutrophils and displays a behavior similar to opsonin. Other bovine peptides like β-casein f(63–680) and β-casein f(191–193) may influence *in vitro* phagocytic drugs in human beings which are being used in immunotherapy of human immunodeficiency which may involve j-casein and α-lactalbumin peptides viral infections. Caseino-macropeptide (CMP) facilitates the development of enteroinfection-inhibiting bifidobacteria and/or lactobacilli.

### 2.4. Antioxidant Peptides

Several milk-derived peptides also act as regulators in the oxidative metabolism, which is essential for cell survival and may also causes oxidative damage through the production of free radicals. As a result of excess free radicals generation, cellular proteins, membrane lipids, DNA and enzymes are damaged, which may lead to several diseases such as rheumatoid arthritis, diabetes, atherosclerosis, and oxidative DNA damage may also cause cancer [79,80,81]. Milk-derived peptides comprise of five to 11 sequentially distributed hydrophobic amino acids released during the process of hydrolysis via the action of proteolytic enzymes from caseins, soy, and gelatin. These include histidine, tryptophan, tyrosine, and proline [82]. These also acts as scavengers or inhibitors of the generation of free radical species [83,84], and can, in particular, affect the enzymatic and non-enzymatic lipid peroxidation of free radicals released from casein peptides [85,86]. Comprehensive research on these antioxidant peptides has shown the strong antioxidant activity of artificial antioxidants against several oxidation systems. Owing to their potential adverse effects on human metabolism and physiology, their use is limited in several countries; consequently, there is more focus on the development of plant-derived natural antioxidants [87]. Naturally presenting vitamins (C and E), β-carotene, and antioxidative enzyme systems with antioxidant potential, particularly catalase, superoxide dismutase, and glutathione peroxidase are now currently being utilized for the production of natural antioxidative peptides [88].

### 2.5. Opioid Peptides

Opioid peptides are the opioid ligands, encrypted *in vitro* from human and bovine β-casein and generally present in the central nervous system, immune system, endocrine gland, and also in the mammalian gastrointestinal tracts [89]. They come into contact with endogenous and exogenous linkages and antagonistic opioid peptides and may affects the blood pressure, fluctuating body temperature, loss of appetite, sexual behavior as well as changes to the central or peripheral nervous system sexually [90,91]. The development and function of central nervous system cells can be controlled through endogenous opioids while the β-casomorphins may be transported via. neonatal mucosal membranes that regulates thephysiological behavior and may induce causes sleep and calmness in children [92,93]. β-Casomorphine, on the other hand, communicates with opioid receptors present on the serosal side of epithelium membrane and plays a significant role in other biochemical processes such as electrolyte regulation, insulin secretion and food absorption [94]. Opioid antagonists can inhibit enkephalin’s agonistic activity. The two effective antagonistic morphine peptides called casoxin C and serorphine were isolated from the fragments of bovine j-CN receptor and bovine serum albumin [95]. The information from the data of several studies suggested that that the two casoxins (A and B) are the ligands of the opioid receptors with a fairly low antagonistic strength [96]. Casoxins A and B are equivalent to a range of amino acids inside bovine K-casein; casexin A is equally equivalent to k-casein f(35–41) (i.e., YPYGLA) while casexin B is equivalent to k-casein f58–61 (e.g., YPYY). Based on these studies, casxin C may acts as a potent morphine antagonist opoid peptide with a high biological capacity [97]. Recent evidence suggests that casomorphines, such as opioid ligands, act against secretion; improve analgesic activity and endocrine reactions [98,99,100].

## 3. Milk Fat Globule Membranes (MFGM): A Value-Added Product in Milk and Dairy Industry

Due to its complex structure and health-efficient properties, milk fat globule membranes (MFGM) has received considerable attention in recent years [101]. The composition and function of MFGM proteins are essential indicators of dairy’s nutritional origin and can be integrated into a range of health benefits [102,103,104]. The use of MFGM as a nutraceutical relies on its chemical components, modifications in manufacturing and different processed food products. MFGM may be isolated from milk or dairy products, including buttermilk and butter strengthened in MFGM components. The separation of milk into cream and skimmed milk, churning of butter, flavor and milk products texture can be related to milk fat globule surface properties. Isolation and characterization procedures determine the composition of proteins. Hence, appropriate use of MFGM as well as its components could therefore greatly enhance the value added of dairy products [105].

The proteins of the MFGM are 1–4% of the overall milk proteins. The complex peripheral and integral protein system represents the MFGM [106]. They play a significant role in infant metabolism and act as a defense mechanism [107]. The milk fat globule membrane protein characterization with polyacrylamide gel electrophoresis exposed approximately nine polypeptide chains. To date, important functional bioactivities related to the fat membrane protein of milk globules include immune stimulation, antimicrobial and antiviral characteristics. The specific components of bovine dairy fat membrane protein like lactadherin show less bioactivity than the human analogous [108]. Two MFGM glycoproteins, namely, lactadherin and mucin display resistance to pepsin and sustain their biological function even at low gastric pH. The involvement of carbohydrate moieties, which form a glycocalyx around milk fats globules, may be correlated with the resistance of the glycoproteins. Glycocalyx provides a barrier against lipolysis as well as a steric barrier against aggregation and recrystallization [109]. In addition to the antibacterial and anti-inflammatory activities of the MFGM [110] proteins, MFGM phospholipids, particularly, sphingomyelin also have many potential health benefits, like gut protection and colon cancer prevention.

The concentration of neutral lipid content of MFGM varies widely. The most common components of total membrane lipid are glycerides. In comparison with bulk milk triglycerides, triglycerides constitute a greater proportion of long chain fatty acids, generally referred to as high melting triglycerides. In MFGM, free fatty acids like β-carotene, hydrocarbons, and squalene represents lipid fractions. MFGM also consists of membrane phospholipids fragment which constitutes phosphatidylethanolamine, phosphatidylcholine, and sphingomyelin. Thus, this cross-examination offers further bit of knowledge of the complex structure of MFGM proteins, strengthening our understanding of the functional importance of MFGM proteins.

## 4. Factors Affecting Milk Bioactive Peptides Composition and the Variations among Animal Genetics

Biologically-active peptides can be generated from milk proteins by various mechanisms that include the activity of microbial enzymes, proteolytic enzymes, and indigenous enzymes from starter and non-starter cultures that function during milk secretion, storage, processing, and finally digestion. The key indigenous digestive enzymes were identified in caprine and ovine milk [111,112,113,114]. Indigenous enzymes play a significant role in the release and storage of bioactive peptides. Numerous peptides have been detected in goat milk incubated for up to seven days even without protease inhibitor, plasmine played a major role in casein hydrolysis and large quantities of peptides were extracted from β-casein hydrolysis. Around 90% of peptides identified were structurally identical in bovine milk, caprine milk and dairy products with previously mentioned bioactive peptides with encrypted sequences capable of exerting antihypertensive action [115,116], antioxidant activity [86], and ACE-inhibitory action [115,117]. Multifunctional peptides resulting from the activity of peptidases of diverse origin in casein factions had been identified in sheep’s milk. A clear example of a multifunctional peptide is the peptide contributing to ovine f(203–208), as it displays not only antimicrobial behaviors but also strong antihypertensives and antioxidants [116].

The composition of milk protein varies between the major dairy animals. For e.g., sheep milk usually contains more casein, α-lactalbumin, serum albumin, lactoferrin, and β-lactoglobulin, than goats, buffaloes and cows [118]. There are several factors that may influence the composition of milk bioactive peptides, such as, protein composition, ruminants diet, animal genetics, environmental conditions, lactation stage, physiological state of the animal, which ultimately leads to milk yield and composition [119,120,121,122]. The components of milk protein are defined well in the literature, nonetheless, knowledge on how animal genetics, animal nutrition and the distribution of these components are scarcely found in the literature. Diverse technical approaches are being used by the dairy industry to adjust the concentration of different bioactive peptides in milk (addition of protein precursors, modified milk proteins, or processed enzymes) [123]. The preferences of customers are increasingly moving towards fresh food products and less processed packaged foods [124]. Animal genetics and nutrition might be significant assets to collect natural products enriched in bioactive peptides. The most efficient and useful way of increasing the production of the beneficial milk bioactive peptides for the consumption of humans is potentially through animal genetics. The profile and concentration of milk protein distinguished between animal species has been strongly influenced by animal nutrition and animal genetics [119]. Unfortunately, only a few studies expand these relationships to allow the modifications in milk bioactive peptides.

## 5. Chemical Analysis of Individual Components of Milk and Dairy Products

### 5.1. Milk Components Analysis

The majority of the milk-derived proteins is comprised of casein (a major portion) and whey proteins. Caseins are the main proteins found in macromolecular aggregates in bovine and ovine milks. Many bioactive peptides from whey proteins are accessible. A few of the known bioactive peptides produced by whey proteins are albutensin A, α-lactorphine, β-lactotensin, β-lactorphine, lachoferricine, and serorphine. Lactoferrin is a predominant whey protein in human milk and plays a key role in intestinal iron absorption. Progress in the identification and detailed description of milk components is linked with the development of new analytical methods. Developments of analytical technologies like ultracentrifugation, chromatography and electrophoresis, precipitation methods paved the way for major caseins, whey and minor proteins separation [12]. Dairy farmers greatly influence the processing of milk components (e.g., protein, total solids, sugars such as lactose, non-fat solids, etc.). The different classes of milk-derived components with their bioactive potential are displayed in Figure 2. MIR milk analysis proves to be a quick, cost-efficient secondary method of testing, based on these results, the regular milk payments are provided to the dairy farmers. MIR conducts all the milk component assessments for each cow for the genetic selection and decision-making techniques. The accuracy and consistency of tests across large geographical areas are therefore of high economic importance and affects business decisions each day.

#### 5.1.1. Harmonized Methods

In the 1980s and 1990s, the development of internationally harmonized standards for inter-laboratory research for the validation of analytical methods in the area of routine quantitative testing methods that affected the testing of milk components was significant. The recommendations were first accepted by the majority of representatives of the method validation organizations, including from International Association of Official Analytical Chemists (AOACI), the International Organization for Standardization (ISO) [125], and the International Dairy Federation. It was published and updated in 1995 by the AOACI [126]. The guidelines apply to validate the performance of the methods used throughout all analytical chemistry, but have been particularly validated for the development and validation of the analytical processes used in the milk and dairy industry. These regulations also provide framework for designing inter-laboratory studies, minimum equivalents of samples and laboratory materials as well as standard statistical procedures for the removal and calculation of outliers and statistical methods of performance. Statistic indexes (e.g., r and R-value) reported in each approach offer specific validated analyst metrics for the predicted study within the laboratory. The guidelines for agreement based on repetitiveness (i.e., replicates) and the anticipated inter-laboratory agreement where the process is properly implemented (i.e., reproductivity) have also been put on practice.

During the last 25 years, the global analytical community is engaged in developing harmonized policies and procedures in order to guarantee the reliability of analyzed data as ISO described in the Eurachem Guide in 1999 and 2000, followed by a 2003 upgrade [125,127,128]. The methods involve multiple operations requiring professional staff utilization, validated processes, internal quality assurance, research expertise and accreditation. In comparison, scientific result precision is inextricably related to the principles of traceability and uncertainty concepts and vulnerability. In 2003, Eurachem reported that analytical traceability corresponds to the relationship among the between analytical research and any appropriate reference point (accuracy benchmark) utilizing a variety of unbroken measurements, all of which are reported uncertain (variability estimation dependent on the dispersion of tests, including system bias) [128]. In United States, milk components such as total solids, calcium, and fat have an indefined (mixed) formulation for payment purposes. The reference criteria that are used for their analysis are the approved methods of measurements. The exception is lactose, as it can be obtained as a single chemical substance and a stable physical activity standard (lactose monohydrate).

The committee of the USDA Test Procedures focused largely on the implementation of these principles under the Federal Milk Marketing Orders. Between 1989 and 1998, significant initial moves took place. Component testing approaches were specified, and the success of their system reported. The introduction of a competency evaluation system for laboratories run under contract was an expansion of this research In order to ensure the system efficiency requirements might be achieve on an on-going basis, individual milk marketing orders. In 1994, 1997, and 2003, Lynch and coworkers reported the results of the progress of UDSA’s test procedures [20,129,130]. The Commission on Experimental Procedures is already ongoing, establishing a program to trace MIR tests by assigning reference values for milk-based materials with uncertainty figures for both model calibrations and validation of the methods.

#### 5.1.2. Chemical Analysis: Primary Methods

Although validated analytical methods conducted on a limited number of samples are considered old fashioned by others, the findings provide the foundation for high-speed method calibration testing. The accuracy of these preliminary tests or comparison among methods is, therefore, crucial, and since all the MIR milk evaluations rely on their accuracy, it is therefore of the utmost importance, as all milk tests rely on their reliability. The 2000 edition of the official analytical methods released by AOACI incorporates the latest versions of these reference methods. The AOACI method 995.19 is usually considered as a standard method for the extraction of fat. The basis on which protein reference testing is carried out is Kjeldahl nitrogen testing, with crude protein measured by AOACI method 991.22 in the United States and in several other countries, while the crude protein was measured by AOACI 991.20 method. The precise MIR measurements for all milk proteins were identified by Barbano and Lynch when real protein has been used as a guide instead of crude protein, since the MIR only detects pure protein at conventional wavelengths [131]. The chemical methods for measuring lactose were less important until recently. However, for a minimum solid, MIR payment check or other solids are required, and then appropriate chemical methods of reference are important for lactose and total solids. Polarimetry lactose calculation is the standard technique, but the AOACI 896.01 procedure is very complicated and usually does not work at a comparable level. Spectrophotometric enzyme methods like the 984.15 AOACI methods have become more common in lactose measurements over the last 25 years. Lynch et al. reported that the lactose enzyme process had been continued to improve by converting the process into weight rather than volume, and research continued to improve the method’s performance [20]. The standard method for measuring total solids is the AOACI drying method 990.20. In the last 25 years, the fat extraction process was greatly improved which includes, utilizing colour indicator for better removal of fat layers without contamination by the analyst; water addition to change interface levels to allow the fat layer removal; extraction inclusion; which an analyst should be able to achieve performance in the inter-laboratory studies [132].

One of the main advances made in the Kjeldahl process was the replacement of a mercury catalyst with an increased concentration of copper sulfate that enables an equivalent or better recovery of nitrogen [133]. Another breakthrough was the invention of sample preparation technique to evaluate true protein directly through Kjeldahl, rather than measuring the difference between different quantities of total nitrogen and non-protein [134]. In this way, Kjeldahl method used as an approach of true protein measurements for the testing of MIR milk analyzers and was adopted by the USDA and published in the Federal Register in 1999. France was the first country to use true protein as a reference for MIR calibration. It is becoming more common in the today’s world to use true proteins for the testing of milk payment and the record keeping of milk production. Barbano and co-workers carried out inter-laboratory experiments in 1990 and 1991 and recorded the efficiency of the Kjeldahl method for crude protein and true (direct and indirect) proteins [133,134]. Similar results for the Kjeldahl method for non-protein nitrogen system were provided in 1991 and 1998, in documenting the success of Kjeldahl’s direct and indirect methods for evaluating the milk casein content [134,135]. In 1999, Lynch and Barbano [136] released a problem-solving guide to help laboratories with improved Kjeldahl’s productivity for the dairy products testing process. Clark et al. and Lynch et al. [130,137] reported significant changes to the complete solid process made over the last 25 years: regular use of the weighing plate (allowing the pan to spread milk equally over the bottom of the container); sample size standardization, time and drying temperatures; start of a dryer from a hot oven.

#### 5.1.3. Electronic Analysis: Secondary Method

Electronic analysis requires a sample collection that optimizes reference values. The first commercially popular electronic tester system for fat came in the 1960s, the electronic milk measuring instruments focused on the light dispersion, known as Milko Testers. Over the last 25 years the design and quality of the MIR evaluation optical table, computer and electronics has continued to evolve and replaced the Milko Tester almost entirely. The MIR system tests infrared absorption of light at traditional milk portion testing vibration frequencies reflecting essential chemical bonds in the MIR spectrum. Total solids are projections based on fat, protein and lactose levels and regression association measures for the total solids as calculated by the oven-drying technique. This is the MIR calculation for the total solids. In order to quantify the fat, sugar, and lactose, a wavelength band is chosen by the MIR-based instruments using a set of optical filters (reference and sample). MIR instrument for the first generation used the carbonyl extension (also called fat A), while carbon dioxide (sometimes called carbon dioxide), as well as the carbonyl stretch, were used by the second generation. Today, optical filter instruments are still on the market; however, Fourier’s more advanced infrared transforming devices using inter-programming are now available to produce full spectral data [138].

While Fourier transform infrared can collect additional spectral data and should potentially make for more accurate tests, this additional information is still to be shown to boost the performance of the method under conditions of realistic milk payments testing successfully. For the testing of milk from cows, the measurement for milk urea at other wavelengths has been made possible, and this information is useful for assessing the effectiveness of dairy cows’ dietary use of nitrogen. When testing milk matrices that include other products, additional spectral information on the ingredients (e.g., ice-cream blend) may compensate for non-milk-based interferences. Individual laboratories are not able to continuously produce their calibration samples and use chemistry. In the future, integrated calibration systems will be built which will accurately design and distribute costs over multiple calibration instruments. Also, coordinated multi-lab testing of common references to calibration and validation samples will also make all samples more reliable and validation of reference values for milk payment testing and process control methods as well as related unsafely estimates.

### 5.2. Dairy Product Composition Analysis

The dairy food processing sectors have undergone massive consolidation over the last 25 years to achieve manufacturing productivity gains and improved marketing and sales. The number of manufacturing plants has declined while each remaining plant’s output capacity has increased considerably. Also comparatively small control system errors are, therefore, significant in large-scale manufacturing plants in the current scenario. Even small errors in process control are thus significant in today’s large factories. The manufacturing quantity of the product per hour is enormous in large plants and process monitoring is extremely crucial. Even though precise, chemical approaches are appropriate for calibration tests comparisons, the efficiency of high-speed system management decision-making needs to be checked. Inline MIR control hardware was developed through the design of electronics and equipment for the real-time standardization of milk composition in the cheese industry [139]. Inline MIR and near-infrared milk (NIR) assays has been generally used for manufacturing and transportation of liquid milk and dairy products along with their composition control (e.g., milk for the production of cheese, liquid milk,). For heavy goods such as milk, though, the case is more complicated. Near-infrared reflectance methods have fulfilled these needs. While NIR machines need calibration each time, a large sample size of the particular product are typically needed for calibration, the analysis speed is crucial for daily basis in comparison with traditional chemical analysis. For process management decisions that have a significant effect on product output, longevity or quality, the product composition has been reduce to a minimum for the better nutritional value.

#### 5.2.1. Sensory Analysis

The historical basis was the milk product rating and scorecard method of the American Dairy Science Association (ADSA). However, researchers and drug producers did not use these approaches very much in sensory tests of dairy products. In 1981, Bodyfelt suggested that the ADSA score card was arbitrary, although it is an important factor in the dairy industry, particularly in recognizing defects.

#### 5.2.2. Instrumental Analysis

A relatively high-intensity characterization of positive product attributes or the detection of interactions could not be instrumental [140]. In the 1970s, a quantitative empirical description was established. Lawless and Heymann [141] addressed QDA as a quantitative tool for data generation which was more expensive than a scorecard solution but which is ideal for mathematical analysis and also of interest to study, growth and basic sensory science. A collection of related goods with the form of the shift are given for a specific kind of commodity (10 to 12). Features that are expected to exist in the population. Panellists and a sensory facilitator are creating a lexicon of descriptive words, enabling panellists to identify the attribute discrepancy between measurements. In the 1970s, a descriptive quantitative analysis has been conducted. In 1999, Lawless and Heymann addressed QDA as a descriptive tool, offering information that is more costly than the scorecard approach and thus of interest for analysis. A collection of related goods with the form of difference are presented to a panellist (10 to 12) for a specific product category possibly observed characteristics in the community. Panellists build a lexicon of descriptive words, along with a leading sensory scientist, enabling panellists to explain the difference between sample characteristics. Where appropriate, study groups with a variety of intensities for each particular function are part of the preparation to allow panellists to make flexible decisions on the level. In a particular research methodology, the certified table is then used to collect data. Data for multiple attributes may be analyzed in spider plots for a fingerprint of the different characteristics of a product. The core elements that distinguish sample populations can be classified by the key components analysis. This method is an example for sensory cheese processing [142].

Through the rapid development and interplay of instrumental approaches through olfactory sensory assessment, sensory research and chemical analysis have taken on a new dimension. Acree et al. [143] announced a reliable new instrument for sensory science in developing the gas chromatography (GC) methods for classifying and calculating the fragrance composition of food products. For instance, hundreds of various blood substances can be present in a meal, which can be extracted by GC and detected; however, only a small proportion of these can cause a sensory human reaction. Splitters and conditioners for column effluents have allowed individuals to allocate a specific term to the retention score to quantify each pit by smelling from the eluent obtained from chromatogram, and indicating, in some cases, the intensity or the period of aroma persistence. Therefore, the peak that has the observable taste from liquid Whey can be separated and analyzed by MS to evaluate their composition and recognition [144]. Carpino et al. [145] used this approach to identify and link specific compounds with QDA analysis results. Specific compounds of certain cheese types have been indicated as a consequence of the intake of local drilling and could be used to certify the sources of protected varieties of cheese as biomarkers.

## 6. Advanced Analytical Techniques of Milk and Dairy Products Analysis

The checking of consistency of food items, like, milk, fish or beef, is essential for the authenticity labelling and appraisal and is also critical in order to avoid unfair pricing and protect consumers from misleading practices widely practiced in the food industry [146]. Most authenticity recognition methodologies for dairy products are based on the major analysis of milk proteins [147]. There has been ever-increasing awareness about the phenomenal diversity of nutrients in milk as well as how these milk components provide several types of crucial bioactivities well beyond just providing nutrition. This focus on the nutritional diversity of milk has been assisted by advancements such as revised analytical techniques that facilitated lower detection levels, the employment of a range of –omic techniques, use of a wide spectrum of *in vitro* and *in vivo* models, and leading knowledge of human milk composition and also other kinds of milk of other breeds. Nevertheless, research is still in its early stages on specific target techniques for the identification and quantitative analysis of these peptides. This information is necessary to increase the chemometric and metabolic characteristics and their subsequent use in the functional food and nutraceutical based industries.

The physical/microbiological/chemical examination of dairy products is a prerequisite tool in evaluating and monitoring the quality of milk and its by-products (raw and processed) to determine the composition, texture, product standardization for their production on a real-time basis [1]. Numerous sophisticated instrumental technique, able to evaluate major and minor components, contaminations and other chemical-induced changes in dairy products processing have been developed to get rid of these problems. They require considerably less time to scrutinize the product and come up with precise analytical results. Success in recognizing milk components and outlining their detailed physical and chemical properties is linked to the development of new analytical techniques [144,145]. Several advances in analytical equipment have been made over the past 100 years, including amino acid sequencing, analytical centrifugation, thin-layer chromatography, rheology, electrophoresis, genomic approaches, dynamic light scattering, neutron and X-ray scattering, GC, electron microscopy, and MS, and many more [143,148,149].

At time a bioactive fraction is attained, further characterization can be conducted after identifying its peptide sequences therein. This customarily comprises of utilizing front end seperative techniques [e.g., capillary electrophoresis (CE) or liquid chromatography (LC)] in combination with MS [150,151]. Since initial hydrolysate is more complex than the obtained fraction (from peptide composition outlook) therefore carrying out MS characterization on the latter is a way to check the peptide candidates liable for distinct bioactive characteristics [152]. Collateral researches may then be executed for added *in vitro* assessment of chosen sequences using synthetic peptides for their bioactive properties.

Techniques like membrane ultra- and nanofiltration can also be applied for the purpose of peptide extraction and purification with additional benefit that no chemical agent is required for doing the same [153]. The series of steps involved are simple to scale-up and can be coupled with other operations. In order to conduct protein hydrolysis and peptide separation concurrently, a membrane reactor can be harnessed to permit a continual process [11,154]. For separating peptides from both fermented and casein-suspensions, ultrafiltration has been widely used where a cut-off optimal value usually ranged from 3 to 14 kDa probably depends on the targeted peptides [7,155,156]. However, the purpose can be served better with the applications of high end and high throughput techniques like MS and chromatographic techniques (in isolation or combination) when a resolution of greater magnitude is required.

The emerging field of peptidomics and its various tools such as chromatography, mass spectrometry, and bioinformatics allow to scrutinize the heterogeneity of peptides more rapidly and accurately. It is now viable to visualize protein regions that are preferentially hydrolyzed with the advent of different computational methods including PCA, MLR, ANNs, and PLS-DA [150,157,158]. Howbeit, an enormous dataset is generated by peptidomics that requires statistical tools to be implemented for dealing with thousands of peptides spotted in a single sample [159]. The gathered data can then be used to construct databases [160], particularly, for milk proteins which have been comprehensively surveyed. Peptidomics is used not only to study the digestion of proteins-it has many applications in dairy science. Dairy scientists employ peptidomics to identify peptides in dairy products, including cheeses [161,162,163,164], yoghurts [165], and kefir [150]. These peptidomics studies revealed that differences in starting materials, production techniques, and storage time resulted in major variation in the peptide profile. For example, peptidomics analysis of lactose-hydrolyzed ultra-high temperature heat-treated milk revealed that residual proteases in the lactase preparation caused protein degradation that led to bitter flavor development with storage time [166,167,168]. Peptidomics can also be used to monitor inter-individual divergence among cows [148] and as a tag for their health status [169]. Recently, a combination of CE, LC, like reversed-phase high-performance liquid chromatography (RP-HPLC), MS, and hydrophilic interacting liquid chromatography (HILIC) have been found to reveal bioactive peptides and their sequences in homogenized milk and human samples like sera, urine, and other intestinal substances [160,170,171].

### 6.1. Spectroscopic Detection

Spectroscopy is the study of the electromagnetic radiation or ultrasound waves in physical systems in which they are transmitted or generated. Several spectroscopic techniques such as low resolution nuclear magnetic resonance spectroscopy, UV-VIS spectroscopy, infrared spectroscopy, ultrasound spectroscopy, microwave spectroscopy, atomic spectroscopy and microwave spectroscopy are currently being utilized for examination of milk and dairy products. Infrared spectroscopy is the most effective and commonly used analytical technique in the milk industry. The technique is based on the absorption of radiation by the sample and is defined by the light’s wavelength, normally mid-infrared and near-infrared. It provides simultaneous detection of all major milk constituents in a simple, fast, precise and eco-friendly manner. During the processing of milk powder, the moisture and water content can be controlled directly in the drying chamber and also the protein and fat content.

Besides chromatographic techniques, also spectroscopic methods are used to classify food-protein hydrolysates. Infrared (IR) spectroscopy is based on radiation absorption in the infrared region due to fluctuations in a molecule between atoms and thus provides information on the chemical composition and conformational structure of food components. The fingerprint area of the IR spectrum, which is the area from 1800 to 800 cm^−1^, is also an instrumental component to analyze proteins and related materials since this is the range absorbed by the amide group bonds (C=O, N-H, and C-N).

Raman spectroscopy is a complementary technique to IR, which also measures vibrational levels of energy associated with the deformation or bending of bonds, although it depends, in contrast to IR, on polarisability changes, mainly on non-polar groups. In this case, a laser in the UV, visible or near IR region excites the sample. Following a similar approach, PLS regression analysis linked FT-Raman spectra to peptide bitterness [172] via FT-Raman spectroscopy, cationic interactions are investigated in peptides, such as discrete phospholipid membrane structures of lactoferricin [173].

Because of its high molecular environment sensitivity fluorescence spectroscopy can provide useful information about small protein and lipid structure changes. The three aromatic amino acids tryptophan, tyrosine and phenylalanine, are inherently involved in milk protein fluorescence. Tryptophan dominates the fluorescence emissions of these three amino acids and provides information about the protein structure, e.g., it is observed that tryptophan fluorescence switches to longer wavelength during cheese maturation. The variation is represented by PCA and is proposed to describe tryptophan’s exposure to the aqueous phase due to proteolysis and increased pH levels [174]. Front-face spectroscopy fluorescence is also used to research the relationships of the β-Ig bovine and various bioactive peptides extracted from β-Ig [175].

Derivative spectrophotometry has been used in several experiments for several purposes, including protein quantification and native and denatured protein analysis. This method can also be used to classify protein hydrolysates to determine exposure. The extent of the aromatic amino acids during protein hydrolysis and the extent of casein hydrolysate encapsulation [176]. This technique can offer some advantages over more conventional ones because of its flexibility, pace and relatively low cost, according to these writers.

### 6.2. Peptidomics Profiling

Proteomics is the large-scale study of protein expression, protein-protein interactions, or posttranslational modifications. Unlike other methodologies that analyze a few proteins at a time, proteomics can analyze thousands of proteins in a single experiment. This ability to analyze thousands of proteins gives the field of proteomics a unique capability to demonstrate how cells can dynamically respond to changes in their environment. Proteomics is a powerful tool for the identification of proteins and studies their localization, functions, modifications and possible interactions or complexes they can form. Proteomics aims to the analysis of several proteins at a time, unlike traditional identification of one protein. Unlike “the genome” there is no single, static proteome in any organism instead there are dynamic collections of proteins in different cells and tissues that display variations in response to various conditions such as stress or infectious processes. Investigation of similar proteins may reveal differential expression both between and within species. The study of proteome can provide a deeper understanding of the biochemical, physiological aspects of milch animal biology, and it is related to productive aspects and disease conditions.

Use of high throughput techniques such as MS/MS for identification and detection of processed milk products are fast, accurate, modern and highly reliable approaches in current research. The modern MS platforms with high resolution offer an in-depth study of proteomics and peptidomics samples, without prior fractionation of samples. Milk proteins have various structures and may influence their confirmation directly in order to undergo proteolysis. Milk proteins have a diverse range of structural arrangements and conformation which may directly influence their propensity for proteolysis.

Milk-derived peptides provide an attractive target in addition to various health-promoting benefits. In milk, the majority of peptide formation is derived from casein while a minor amount comes from whey protein [169]. Unlike the tightly packed complex globular configurations of α- and β-lactoglobulin and other whey proteins, the basic rheomorphic arrangement of caseins makes them more susceptible to proteolysis [169,177,178,179]. Bhattacharya et al. [1] identified more than 300 β-casein-derived in all the samples. At the same time, s1- and as2-casein derived peptides sequences with lower frequency by liquid chromatographic (LC)-Orbitrap tandem mass spectrometry (MS/MS) with immunomodulatory, antioxidative, anti-hypertensive, anti-thrombotic, opioid and anti-microbial functionalities.

Chromatography cation-exchange permits the separation and quantification of individual amino acids from acid-hydrolyzed peptides. Anti-hypertensive peptides are the most commonly studied bioactive peptides. These peptides may prevent the angiotensin I converting enzyme (ACE) which catalyzes the conversion of angiotensin I to angiotensin II. The development of angiotensin II also diminishes the vasodilatory properties of bradykinine and together with raises blood pressure; ACE also lessen the effects of bradycinin [180]. Several bovine and human caesin-derived peptides have already been identified as ACE inhibitors to date [181].

### 6.3. Chromatography

Peptide purification can be accomplished using different chromatographic methods that include ion-exchange chromatography (IEC) and size-exclusion chromatography (SEC). These methods are immensely selective in nature and provide a high-resolution separation. The core problem regarding their industrial applications is the exorbitant cost associated with these techniques. Moreover, chromatography brings about peptide dilution and agglomeration of solvent wastes [182]. Just like peptidomics, chromatographic techniques also have research applications and are being widely used for same. For example, gel filtration chromatography was used from fermented milk to purify and prepare anti-tumor peptides with the help of *Lb. helveticus* [183]. Owing to its high resolution, this technique of purification is also notably suited for recognition of the specific peptides that are involved in a given bioactivity. Peptide identification is essential to establish that a particular activity is affiliated with a specific BAP. Pharmaceutical applications demands complete identification and sequencing of active peptides [157,160,184]. Profiteering of separation and documentation methods such as chromatographic isolation and electrophoresis have been used for a prolonged period. These techniques have principally been targeted to carefully comprehend either the physical or chemical stabilities, or the bioanalysis of peptides at the highest degree of their sensitivity, by identifying the amino- and carboxy-terminal residues and the amino acid sequences [185,186,187]. For example, structural identification and detection focusing on physical stability of peptides has been mainly achieved through the SEC, by examining the elution volume, time of migration and peptide aggregation at higher rates [188]. RP-HPLC has also been used to detect peptides by studying changes in deamidation, polarity and water repellence characteristics [189].

Currently, in addition to SEC and IEC other techniques, such as solid-phase extraction, HILIC, affinity chromatography, and preparative RP-HPLC [190] are the most employed methods at the laboratory scale. A peptide’s physicochemical properties form the basis for picking a specific chromatographic purification technique to be adopted for the purpose. For example, HILIC represents a versatile and effective alternative that performs better than RP chromatography (RPC) for the study of hydrophilic peptides (polar peptides). On the other hand, IEC separates the peptides on the basis of the charge and therefore offers a different selectivity; nonetheless, its main drawback in a complex mixture is related to separation challenges faced when peptides generate the same charge. In regard to SEC, it has been primarily preferred for everyday and validated analyses owing to its speed and reproducibility, yet challenges to interface SEC along MS exists. Moreover, the noteworthy advancements in resolution, sensitivity, and throughput of SEC technique in view of using smaller particle size have complementedits potential. In order to replace batch methods of salting-out or solvent extraction for the purpose of bioactive peptides isolation and purification, a lot of effort has been placed to establish selective column chromatography. Betterment made in this direction would ameliorate BP recovery to great extent. Online multidimensional chromatographic systems surfacing nowadays also have significant potential to conduct separation of peptides in improved manner and has lead to multiplied number of identified sequences of peptides.

### 6.4. Mass Spectrometry

Innovations and substantial refinements have led to the evolution of many diversified analytical techniques in the domain of analytical chemistry for the isolation and purification of low-concentrated compounds. One such high-resolution technique is mass spectrometry (MS) that is capable of performing analysis of intensely complex mixtures while detecting several distinctive categories of analytes in a wide array of concentrations [191]. The data produced through MS has to be analyzed using variant algorithms, software and databases. These various analysis platforms grant processing of enormous data in automated and inclusive manner. The software used for identification and quantification depends on the instrument used as the output file type differs among instruments from different vendors and are either furnished with the instrument itself or are provided by the suppliers (Figure 3).

In the 1990s, MS, coupled with liquid chromatography, was widely used for the study of dairy components. One of the first novel insights from this technique was those lactose molecules could attach to milk proteins via the Maillard reaction (called lactosylation) during the prolonged storage of milk powders. It has become a key analytical tool for identifying bioactive peptides, minor components, and nature of whey protein aggregates; confirming levels of phosphorylation in the CN, and countless other applications in dairy chemistry. Propitious studies based on LC-MS have made it possible to reveal that certain BAPs from intact or hydrolyzed proteins are present within the human body after characterizing fluids from the human gastrointestinal tract [98] and under circulation [192,193,194]. Although, supplementary quantitative studies are much required to validate that these BAPs exist in ample quantities to have their biological effects.

MS-based techniques have generated great interest in the latest scenario aiming for structure-informed identification and quantification of biological peptides. This recent interest is attributed to its potential to tackle all the issues linked with complex biological systems with assured reproducibility alongside sensitivity, specificity, and high-throughput analytical proficiency [149,150,195]. Several distinct MS instruments are appropriate for peptidomics, including Orbitrap, Q-TOF, and others [150]. Label-free relative quantification can be accomplished using peptide signal intensities. The ion-signal intensity procedure uses extracted chromatographic areas to compare peptide abundances across samples. MS is not only appropriate for small sized molecules, but also to a wide range of compounds [196]. It has made significant advances with the application of ionization techniques, such as laser matrix-assisted desorption/ionization (MALDI), electrical pulverization (ESI), and desorption electrospray ionization (DESI) in tandem [149,197], in contrast to LC [198]. MS can help to differentiate between various peptide analogs, according to molecular weight and differential spectrum, from either MALDI or ESI spectra [196]. The most prevalent method of analyzing the MS is bidirectional: online and offline. In combination with atmospheric pressure ionization (API), online experiments include LC, but offline methods are preferred to implement the MALDI technique [196]. MS-based peptide quantification technological advances have indeed influenced several other physical perspectives, either in isolation or in association with other chromatographic and electrophoretic techniques [149]. The superiority of MS in structure-based peptide identification [169], using unconventional methods such as MS-based ion mobility and imaging [150] has been reported in detail. Both these approaches are distinctive, since they greatly contribute to the achievement of tissue analyses with spatial and temporal resolutions at the molecular level and with various isolation levels. The greatest achievement in MS, however, is certainly evidenced when used in combination with other high throughput analytical tools, as in the case of chromatographic and electrophoretic techniques. For example, on the basis of characterization for biologically active milk peptides, the threshold MS efficiency was reported to be the combination of SRM/MRM and ESI-QqQ (triple quadrupole) [199]. In addition, the LC-MALDI-SRM MS had been used for the quantification of peptides and final results were in accordance to that of LC-ESI-SRM MS [200].

Relatively high mass resolution and precise measurements of mass to charge ratios offer greater accuracy and a less false-positive ratios for the similar number of possible positive values in the pursuit of peptide tandem mass repository. The exploitation of high-resolution MS is a requirement for peptide synthesis and the subsequent validation process. Nevertheless, short peptide sequences create additional issues regarding ion transfer of low molecular mass and over fragmentation. Conventional MS strategies for short peptide sequence analysis involve the use of multiple reaction monitoring (MRM), sequence tags, and chemical derivatization for the analysis of theses peptides. Lahrichi et al. [152] recently introduced an LC-MS/MS system that has been based on an MRM strategy for 117 peptides with 2–4 amino acids. Approximately 60% were unique, although many were isobaric and co-eluting.

However, before MS/MS analysis were carried out, these methods allowed further and tedious steps; therefore, alternative MS-based methods were suggested [201]. Nanostructure laser desorption/ionization (NALDI) is among them [202] that recruited peptides derived from bovine milk and colostrum for identifying low molecular weight nanostructures. Apart from MALDI, NALDI is matrix-free and has no matrix background below *m*/*z* 700; as a result, mass spectra of small peptides with high susceptibility and low chemical background are obtained. By integrating chromatographic information with the MS analysis, a synergistic effect can be achieved. Le Maux et al. gave a valuable example in this regard [203]. HILIC separation has been used to distinguish peptides with homologous sequences by connecting retention times with a certain algorithm with an apparent hydrophilicity and peptide size. An important contribution, for example, is simple ultra-filtration with 5 kDa MWCO, which is followed by nano-UPLC analysis and high-resolve MS/MS which allows 17 short peptide sequences to be analyzed without a chemical derivation [201].

### 6.5. Electrophoresis

A systematic research approach was proposed based on *p*-casein isoelectric concentration and HPLC [204]. However, the findings were approximate as the measured proportions of bovine milk in cheese mixture were highly dependent on the casein composition of the milk used for cheese processing. Techniques of quantification of dairy species are dependent on the whey protein fraction, however, the inadequate fraction in whey protein is more prone to heating as compared to the casein fraction. These approaches will thus produce false negatives in the processing of cheese when sterilized or powdered milk is used. Excessive proteolysis can also be undermined for quantification during cheese ripening. To assess the adulteration of goat milk products, Cartoniet et al. [205] established capillary zone electrophoresis. The identification and quantification of the milk from cows were based on the existence by the relative calibration curve of the different whey proteins. The minimum measurable quantity of cow’s milk was 2% in different milk varieties and 4% in cheese. Only one of the two varieties of milk used have been studied as regards disadvantages due to genetic variation and potential heat treatment. Molina et al. [206] conducted the study by using capillary electrophoresis of goats, sheep and goat milk mixtures. A multivariate regression analysis has been used to quantify the number of adulterated samples.

### 6.6. Enzyme-Linked Immunosorbent Assay (ELISA)

Enzyme-related immunosorbent assay (ELISA) is the most common method of analysis of milk components based on immunoassay and it has advantages of low cost, high specificity, and quick use. It is easy to handle, reliable, fast and automatic [207,208]. Principally, the presence of unrecorded milk in individual animals can be observed by two simple ELISA methods: sandwich ELISA and indirect ELISA with their different variants. The development and practical use of immunoenzymatic approaches depend primarily on the identification of immunogens, experimental models, immune-based methods, consistency of antibodies being utilized or probably used, accuracy and reliability of the proof mechanism [43,209]. Cows and goats’ milk can be tested in milk mixtures using polyclonal and monoclonal antibodies generated to combat proteins, caseins or whey proteins. Milk proteins are short-string peptides. The caseins that form the more significant part of the protein fraction have advantages that under high-temperature conditions, they are more or less stable. Therefore, they can be successfully used as the primary antigens of milk and dairy products in the thermal treatment (pasteurization). Their major drawback is their low immunogenicity and their greater prostheolytic degradation sensitivities. In comparison to casein, whey proteins are much weaker immunogens and proteolytically degradable even in minimum concentrations. Whey proteins are less immune to high temperatures. There are actually a limited number of ELISA studies with a very high ability to detect the contaminants in the heat treated milk [210].

The identification and quantitative analysis of bovine milk adulteration in goats were conducted by an indirect ELISA. The polyclonal antibodies were modified for research purposes by combining with goat’s milk. Absorption at 450 nm in indirect ELISA revealed a clear association with adulterated bovine milk concentration in the range of 4% to 5%. The identification mark for mixed milk samples was 4%. The study was easily reproducible with intra- and inter-assay variance coefficients of less than 5%. Therefore, ELISA can be successfully employed to assess milk sample adulteration and is ideal for creating a package in the daily milk inspection [146]. Zelenáková and Golian [210] describe a technique to evaluate the fraction of goat’s milk in which the authors focused on laboratory testing and quality parameters assessment of the ELISA research. Goat milk was measured and quantified by the existence of standard immunoglobulins (IgG). In order to determine the amount of Igs, around 43 samples from 43 different varieties of sheep and goat dairy blends were tested, and 86 values were determined. By using the specification and a sensitive ELISA test, it has been verified that a standard curve with a limited detector range affects the efficiency of adulterated milk detection. It has been estimated that heating the milk (71.7–77 °C for 20 s) affected the adulteration detection.

The manufacturing companies should confirm that milk testing units/laboratories must have analytical technologies in order to decide on any possible fraud. Evaluation of any commercial ELISA system for detection and quantification is very important for the detection of milk and cheese adulteration [211,212]. The high defined concentration levels, where the effects of ELISA are reported worldwide (mg/kg or µg/kg), are also represented in this extensive system. The main benefit is to process large sample pools, create calibration curves and simultaneously measure blind samples on a single microtitration plate, thus eliminating the effect of changing circumstances in the determination process. ELISA is a food allergen sensing technique too as it is appropriately sensitive for the identification of food-allergen contaminants. ELISA may also be manufactured in formats consistent with the food processing industry environment. However, ELISA still has drawbacks which should be appropriately determined and accommodated [213].

### 6.7. Polymerase Chain Reaction (PCR) Based Detection

Molecular biology methods for species identification in foods of animal origin have been used relatively recently. These techniques are used for their advantage in detecting extremely low concentrations of cow’s milk, either 0.1% or 0.5% [214] for quick and sensitive polymerase chain reaction (PCR)-based methods [215]. Moreover, in contrast to protein-based refined approaches, which are not often applicable and have to be carefully chosen, DNA-related methods such as PCR are being widely used for mature cheese [204] and hot milk products [216]. However, the effort needed to use this approach as a quantitative food authentication technique is also minimal. A straightforward approach was described for assessing the incorporation of cow’s milk in goat cheese, even if this method does not allow sample or gel preparation changes [155]. Another method has been developed for measuring the addition of cow’s milk in sheep’s milk with a duplex PCR technique which uses a standard calibration curve for controlling the problems of extracting DNA and gel [217]. Duplex polymerase chain reaction allows partial or even complete identification of substitution of cow’s milk with buffalo cheese, inaccurately called “pure buffalo” mozzarella cheese in some cases. A primer-based method have been introduced for the species-specific, giving rise to amplified fragments of 279 bp (bovine) and 192 bp (buffalo) [207].

Mašková and Paulícková [218] have used a PCR system to detect cow milk in cattle and sheep cheeses. DNA from cheeses has been preserved using the Invisorb Spin Food I isolation package built for animal samples by Invitek Co. (Berlin, Germany). The PCR system employed uses the mitochondrial gene sequence coding cytochrome b unique to mammals. It uses a standard forward-primer and a unique reverse-primer. Following electrophoresis, cow DNA was distinguished by a 274 bp fragment, goat DNA by a 157 bp fragment and sheep DNA by a 331 bp fragment. Models from pure goat’s cheese determined the detection limit for the PCR method described (1%) was defined by the addition of cheese prepared from cow’s milk. For validation purposes, cheese obtained from 17 goats and seven sheep were analyzed. In three types of goat cheese and sheep’s cheese, the presence of undeclared cow milk was found. Mafra et al. [41] detected adulteration of cow’s milk in the sheep’s milk by utilizing mitochondrial 12S rRNA primers. This technique permitted 0.1% of cow’s milk to be detected utilizing a 35-cycle duplex PCR and quantified by PCR with 30-cycles by using uniform calibration curve used for established cheese. Bobková et al. [219] used the PCR detection system to diagnose cow milk adulteration in eight intentionally falsified sheep milk samples from and a detection limit of 0.01% mixing of cow milk was observed.

### 6.8. Role of Bioinformatics When Applied in Conjugation with Peptidomics

The tools applicable for the discovery and identification of bioactive peptides (BPs) arose with the arrival of shotgun proteomic technologies. Shotgun proteomics is a bottom-up approach where a complex protein mixture is digested explicitly to peptides with the help of an enzyme and these peptides are then analyzed by an integration of HPLC with MS to come up with the parent proteins afterwards using a bioinformatics analysis of the experimental spectra. However, challenges associated with the analytical complexity of nutritional proteomics research still persist despite the enhancements achieved in shotgun proteomics analysis. The diverse selectivity and unique characteristics conferred by multiple *in vitro* and *in vivo* proteases for food protein processing and digestion yield extremely complex peptide mixtures along with short peptide sequences [190].

Various in silico approaches can accelerate the identification of peptides present in complex mixtures. Recent reviews have elaborated peptidomics methodologies currently being used for studying BAPs [150,220]. Distinctive in silico tools have been put forth as the quality of peptide fragments may not always be satisfactory to pinpoint the correct amino acid sequence. Also, the adoption of BAP databases is considerable for the identification purpose of peptides with already expressed bioactivities, truncated/precursor peptide sequences or sequences displaying aspects of BAPs [220,221]. A few studies also summarized how computational methods, like QSAR, could be enforced to foresee the bioactivity of peptides spotted in composite samples and achieve a focused identification of BAP sequences spotted by LC-MS [222,223].

### 6.9. Endogenous Bioactive Peptides (BPs) Analysis

The analysis of BPs (which have been extensively studied) that are produced post *in vitro* simulated digestion have served as the ground for methods and applications outlined hitherto. However, the endogenously produced peptides (part of milk and derivative products) are still poorly investigated as they cannot be revealed by *in vitro* digestion approaches. Additional challenges to the ones reviewed above exist when exploring endogenous BPs. First, the enhancements of native peptides in the matrix must be addressed by isolation techniques, and adoption of the formerly explained protocols can address this issue. Moreover, the identification using traditional proteomics approaches is more challenging when addressing the peptide mixtures of partly known origin. Second, some cases require the search to be expanded from a single organism to a genus or family as protein databases are usually inadequate in the case of organisms that are only partially sequenced. Third, the production of BPs by events that are more convoluted than simulated digestion may give results that are not entirely illustrative. Endogenous peptides can be produced by a variety of ways.

The multiple ways of functioning of endogenous proteases (which may, in addition, be present in the organism of concern, such as proteases in mammary glands for milk, or extracted from other species) or entirely unidentified processes (e.g., thermal processing) can distinguish them from the tryptic peptides that are produced in classical proteomics experiments. In the former case, enzyme specificity is absent or lacking, therefore, identification by matching peptide and fragment masses to sequence databases turns out to be complicated. De novo sequencing and protease unspecific database searches are plausible fixes to this problem. The formation or matching of the fragment ion spectrum to the collection of silicon ion fragment lists from protein sequence repositories using predetermined rules such as tryptic cleavage is not important although de novo peptide ion sequencing enables atypical digested peptides to be further identified.

Another possibility of finding atypical peptides is an unspecified database search (namely, a search without any pre-defined digestion enzyme). In such a search, all potential cleavage sites are examined for developing fragment ion lists from protein sequence databases. This approach provides a much broader search space whereas searching with a selected digestion rule generates a manual inspection requirement as the probability of incorrect identifications is enhanced [190,224].

### 6.10. Coupling Different Techniques

An alternative, multifaceted, and less time as well as sample-consuming method used in hypoallergenic infant milk formulas was capillary electrophoresis [225,226]. Generally, in this regard, to provide adequate resolution mono-dimensional (1D) approaches are incapable, whereas, two-dimensional liquid chromatography (2D-LC) methods provided tremendous improvements. Currently the technique that offers the maximal separation efficiency and the most preferred choices for bottom-up proteomics and peptidomics is considered to be 2D-LC coupled to MS/MS. In short, if the entire sample is exposed to two different separation or heart cutting, 2D-LC is complete only if a part of the sample that flows out of the first dimension is forwarded to the second dimension [227]. A good illustration of how 2D-LC workflows could provide a handy input for BP analysis in the segregation of milk-soluble fraction peptides after their expiration date was presented by the work of Sommella et al. [228]. They demonstrated that peak capacity and resolution can be significantly elevated in 2D-LC while working on an online exhaustive LC × UHPLC platform and compared the outputs with a model highly competent 1D separation having same analysis time. Sanchez-Rivera et al. briefly explained other examples using 2D-LC separation in a review [190]. Thus, distinct methods turn into a feasible and alluring substitute for BP screening which directly employed a fusion of chromatographic techniques conjoined with high-resolution MS analysis, bioinformatics, and database searches.

## 7. Potential Roles and Applications for Milk and Dairy Industry Infant Formula Products

Technological advances in recent decades have greatly affected living conditions and the dairy industry. Production of food has transformed from small-scale farming to mechanical processing, transforming food supply, storage, distribution and consumption on a large scale. Infant’s formula is generally used as a nutrient supplements in order to meet the needs of infant and children. The industrial production of baby formulas requires essential nutrients at a degree higher than normal in breast milk, including Cu and Mn. The only way to nutritionize many infants from the first four to six months of their life is to deliver infant formulas. These are crucial for infant’s health during periods of severe shortages and inadequate nutrition, as they will adequately sustain growth and development. These formulations are consumed by many of the infants and marketed in powdered, liquid and liquid ready-to-eat forms. Notwithstanding advancements in the development of formula preparations, there are a few compounds present in human milk, including anti-infection agents (human milk proteins are well-known to promote cell growth and proliferation), enzymes and trophical factors [229].

Milk proteins are widely used in baby food. The composition difference between human and cow milk is now well known and they do not have the same composition, especially regarding proteins and carbohydrates. Moreover, lactoferrin and α-lactalbumin are the major whey proteins in human milk, while β-lactoglobulin does not exist. In addition, caseins predominate in cow milk, while human milk is rich in whey (soluble) proteins [230]. Thus, αs1- and αs2-casein account for more than half of the total caseins in cow’s milk while bovine caseins are even more highly phosphorylated than human milk casein [231]. The hydrolysates in cow milk proteins are perhaps the most utilized protein in infant formulas due to their superior nutritional value [232]. A variety of cow’s milk products (e.g., non-fat milk, casein, casein and whey protein combinations or whey protein concentrate partially hydrolyzed) focus on providing protein for these formulas.

Milk basic formulas are divided into two categories; whey-predominant and casein-predominant formulas. Casein-predominant formulas are essentially diluted bovine skim milk, which contains fat and other nutrients but does not change the casein: whey ratio. New rules and guidelines for infant foods indicate that it is not compulsory to have both the ratios. The minimal and maximum protein must always be adhered to in baby formulas, although it varies on the source of the protein. The difference between these two formulations is their protein source [231]. Infants fed with whey contents showed enhanced plasma threonine concentrations [230,233].

## 8. Knowledge Summary and Future Directions

All the research pertaining to the detection and identification of minute peptides (<4 amino acids) present in multifarious mixtures are in their early stages because of a lack of stringent peptide identification methodologies. Precise amendments like discerned censoring of ions against previously identified sequences of peptides can help overcome the aforementioned issues faced at times of optimization procedures during or after MS analysis. A state-of-the-art genesis in structure-informed peptide identification and quantification methodologies can be guaranteed by added enrichment in the sensitivity and resolving capacity of MS, in conjunction with novel cutting edge ionization techniques. A remarkable challenge is the invention of new technologies that will secure high production with augmented purity in the domains of chromatographic and non-chromatographic separation procedures, respectively. There is a compelling demand to revise the accessories along with the techniques themselves. Modernization of the software for foodomics and peptidomics research and peptide identification is needed. Also, explicit and coherent structure identification in common and especially in synchronization with LC-MS requires significant attention. However, due to the inadequacy of technological advancements, enriched food items and molecular approaches, there is still limited work in this area.

Over the past 100 years, the different components in milk and the primary forms of milk proteins were established with tremendous advances. Moreover, very little or no knowledge of the stability, bioavailability and efficiency of these bioactive peptides leads to a major knowledge gap that hinders a better understanding of their role in human health. At this point, further prerequisites are the establishment of resources to protect/expand the operation of bioactive peptides and encourage their maximum use in food production systems. Analytical development has contributed to the fractionation and characterization of dairy components. Throughout the past 50 years, knowledge of milk factor biosynthesis has progressed rapidly. Milk testing has also been transformed from slow laboratory procedures into quick tests of multiple components which may be carried out on the farm. Improved understanding of the different forms of milk protein has encouraged the commercialization, in dietary uses, of new milk protein ingredients. A continuous focus will be given to understanding of the biochemical functions of milk ingredients and their dietary implications by using a variety of powerful tools like -omics, cell models, gut microbiome research and imaging. Milk monitoring has grown from gross compositional monitoring for regulatory reasons or farm remuneration to a range of assessments for uses such as agricultural management and animal welfare. Traditionally, tests were conducted in large centralized laboratories and gradually switched to the field itself. Approaches like ELISA are already being used to show the possible occurrence of Johne’s disease in cow’s milk, and these approaches also provide farmers with evidence to take steps to reduce the incidence of this disease. PCR, which is currently in use, is another example of a diagnostic testing aid used to recognize mastitis-causative bacteria. These data have helped organizations identify chronic pathogen shedders and contribute to making management decisions for removing these particular animals and reducing SCC levels. More than 30 tests (composition and indicator) can currently be performed on milk samples, and the number of valuable tests is expected to continue to grow. Increasing technology will also include robotic milking systems, making it easier to collect and test milk samples. In the future, further field milk studies will occur as this provides farmers with timely data for management decisions and the production of milk. Milk is easier to process than blood samples or other biological materials.

The introduction of innovative facilities including is an absolute requirement for the development of approaches, such as proteomics, recombinant enzymes and microbial fermentation to study and improve the metabolic and health consequences of the various roles of bioactive peptides throughout the expression of genes. Consequently, the formulation of products incorporating bioactive peptides should examine the allergenicity, toxicity and stability of the affected metabolic functions during gastrointestinal digestion. The implementation of integrated research platforms is still necessary for interdisciplinary research to clarify the role and mechanism of milk-derived bioactive peptides. In addition, before formulations are used as chemotherapeutic agents or tested directly for viable conditions, the preliminary positive effects of milk derivative products on target diseases must be considered carefully. Despite considerable progress in the isolation, purification and assessment of bioactivities of BP from various natural sources, several hurdles still remain to be overcome, particularly technological advancements to produce them on a broad scale without losing activity. In conclusion, milk-derived bioactive peptides offer substantial future prospects for product development to support health, with their multifunctional assets.

## 9. Conclusions

Bioactive peptides have sparked researcher’s attention as health-promoting functional foods. Nonetheless, research in this field is constrained by the absence of innovative technology, purified materials and molecular techniques. Focusing on the construction of new techniques, including proteomic approaches, advanced and recombinant enzyme technology and microbial fermentation, to study and to standardize the nutrition and dietary effects of various bioactive peptides in genetic expression. Therefore, potential health promoting effects of milk and dairy bioactive peptides on opportunistic infections must be carefully monitored before being using them as chemotherapeutic agents or trying to use in their viable state. Therefore, it is necessary to separate their versatile functional characteristics into food and pharmaceutical products, isolation and identification of these peptides and their related pharmaco-dynamic parameters. Scientific research and industrial growth, in pursuit of suitable novel bioactive peptides, assures the formulation of numerous drugs and functional food products with health-aids. Numerous computational tools are theoretically necessary for milk authentication use. They differ in complexity and expense, and each of these factors may affect food safety laboratories taking up this testing. Tests such as the ones described above, many of which are commonly used in research areas, could be used for market use with the possibility of improved oversight of food products. Until this happens, however, strict validation procedures are required. In addition, the efficiency and reproductivity of these techniques is still under review for general use, and some of the presented procedures maybe applicable in practice. The combination of highly efficient chromatographic techniques, MS/MS, ELISAs and PCR methods will ensure that food analysts can indeed gain sufficient proof to implement European Commission legislation and regulate adulteration in dairy products.

## Figures and Tables

**Figure 1 molecules-25-03328-f001:**
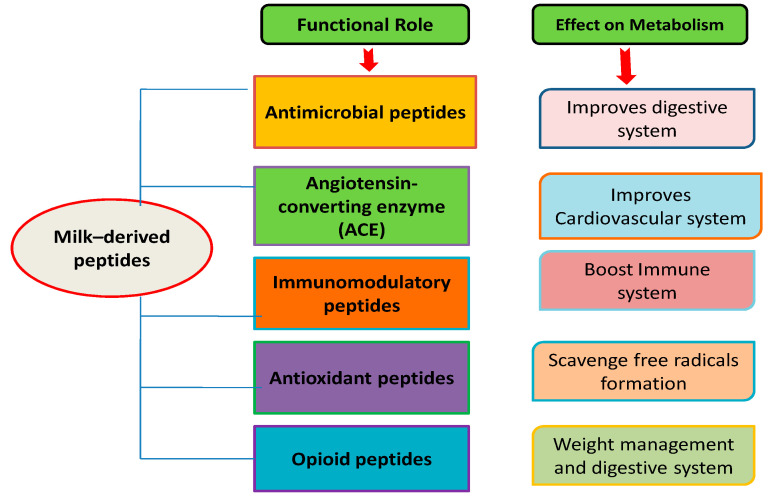
Different milk-derived bioactive peptides in human metabolism.

**Figure 2 molecules-25-03328-f002:**
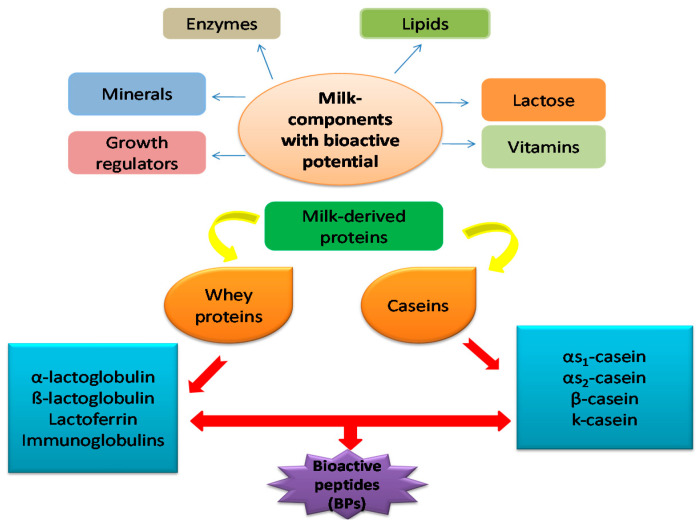
Major bioactive components of milk with health-aid properties.

**Figure 3 molecules-25-03328-f003:**
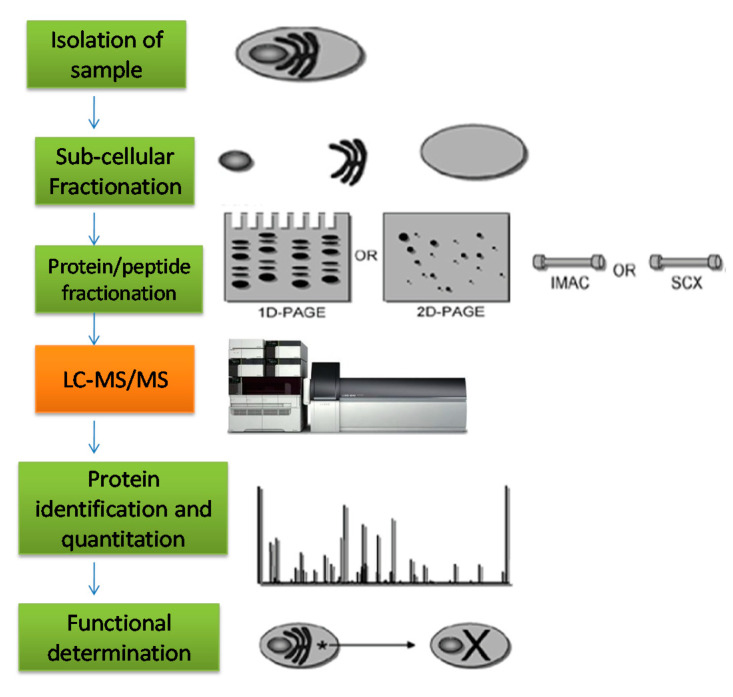
Representation of sample preparation, identification and detection of milk components by LC-MS/MS.

**Table 1 molecules-25-03328-t001:** Techniques used in the identification and detection of bioactive peptides in milk and dairy products.

Analytical Techniques		Chemometric Techniques	Application	References
Electrophoretic Methods	Urea-PAGE	-	Evaluation of changes upon proteolysis in cheese	[21]
	Urea-PAGE of casein	-	Determination of concentration of peptides and age-related differences	[22]
**Chromatographic and Spectrophotometric Techniques**				
Mass Spectrometry	2-DE; MALDI-TOF; ESI-IT	In-gel digestion	Polymorphism of goatαs1-casein	[23]
	RP-HPLC < 1000 Da	Analysis of volatile compounds	Evaluation of changes upon proteolysis in cheese	[21]
		Sensory analysis	Differences among caseins peptides and sensory attributes	[22]
	NanoESI-QTOF	In-capillary tryptic hydrolysis	Characterization of elephant milk proteins	[24]
	MALDI-TOF (reflectron), HPLC-ESI-IT	Tryptic digestion	Identification of truncated goat	[25]
	MALDI-TOF (reflectron), HPLC-ESI-IT	Tryptic digestion	Identification of truncated forms of goat αs2-CN A and E	[26]
	ESI-QqQ	Offline RP-HPLC	Degree of glycosylation and phosphorylation of ovine and caprine CMP	[27,28]
	2-DE, immunoblotting, MALDI-TOF, ESI-QqQ	In-gel digestion immunoblotting	Phosphorylation and glycosylation of ovine caseins	[29]
	1-DE and 2-DE, ESI-IT	Enzymatic digestion	Phosphorylation, glycosylation, and genetic variants of κ-casein	[30]
	2-D LC-nanoESI-IT	Shotgun proteomics	Identification of minor human milk proteins	[31]
	2-DE, HPLC-QTOF	In-gel digestion	Characterization of minor whey proteins	[32]
	HPLC-QTOF	In-gel digestion	Lactosylation of β-Lg, α-La, and αs2-CN in infant formula	[33]
	HPLC-ESI-QqQ	Tryptic digest in solution	Lactosylation of β-Lg	[34]
	MALDI-TOF	-	Degradation of αs1-CN f(1–23) by bacterial amino and endopeptidases	[35]
	MALDI-TOF (reflectron)		Specificity of peptidases from *Lactobacillus helveticus*	[36]
Polymerase Chain reaction (PCR)	PCR-RFLP	Bovine DNA in cheese	Ovine and caprine cheese samples	[37]
		Bovine DNA in cheese	Commercial mozzarella and feta cheese samples	[38]
		Bovine DNA in cheese	Experimental binary mixtures of bovine milk with ovine, caprine, and buffalo milk	[39]
	Duplex PCR	Simultaneous detection of bovine and buffalo DNA in cheese and milk	Experimental mixtures of bovine and buffalo milk and commercial buffalo Mozzarella samples	[40]
		Commercial cheese samples	Simultaneous detection of bovine and caprine DNA in cheese	[41]
	RT-PCR	Bovine DNA in cheese	Experimental and commercial mozzarella cheese	[42]
	ELISA	Identification of immunogens	Evaluation the fraction of goat’s milk for quantification and the existence of standard immunoglobulins (IgG)	[43]

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
