# Peer review of "Identification and Detection of Bioactive Peptides in Milk and Dairy Products: Remarks about Agro-Foods"

_molecules, 2020, doi:10.3390/molecules25153328_

Round 1
Reviewer 1 Report
This is a vast overview of an area of growing interest in the development of milk based specialized foods and functional foods. However, certain concerns are noted as described below.
Major Comments:
- Many of the references are too old/outdated, with large numbers dating from the 1980s and 1990s. Please focus on more recent trends and developments, especially studies published in the last 10-15 years. This is critical for the review to be focused on the current knowledge base and future directions.
- The chapter on milk components is poorly organized. The key parts of milk are: water, proteins (the focus of this review), lactose and fat. Of the protein component, over half is casein in bovine milk. The rest are comprised of various whey proteins. Many methods separate the casein from whey and treat these components separately to yield different peptides. This should be clearly articulated in the article (as shown in figure 2).
- It would be preferable to have a paragraph or two discussing the potential roles and applications for the infant formula industry, which is a growing sector in many parts of the world.
- Milk fat globule membranes (MFGM) is another component, with a mix of proteins and lipids, that is of potential use as a value-added product. It is suggested to discuss this in a separate paragraph.
- The section on analytical methods could be condensed and edited for length and clarity. Proper references should take place of excessive details.
- It will be preferable to have more details on future directions. Which areas and which food products are most likely to undergo greatest innovations in the next decade?
Specific Comments:
- It is surprising that milk/dairy comprises 25-30% of diet (as stated in line 141). Please provide further references and discuss what populations these figures are based on. It seems extremely high.
- Anti-hypertensive: Please discuss the roles of milk derived tripeptides VPP and IPP which have been commercialized and tested in clinical studies for their BP lowering properties.
- Anti-diabetic: Many milk-based peptides (including IPP, VPP) have shown promise as insulin mimetic and potential anti-diabetic agents. These should be described under a separate chapter (instead of brief mention under opioid peptides, since the anti-diabetic effect is not necessarily mediated through opioid receptors).
- The introduction should refer to several recent reviews on food derived bioactive peptides to provide a better overview to the readers.
Author Response
Pointwise Answers to the reviewer’s comments
|
S.No |
Comment |
Reply |
|
|
Reviewer #1: |
|
|
1 |
Many of the references are too old/outdated, with large numbers dating from the 1980s and 1990s. Please focus on more recent trends and developments, especially studies published in the last 10-15 years. This is critical for the review to be focused on the current knowledge base and future directions. |
To make the review more comprehensive, all the research studies have been incorporated. Still (if suggested), will remove the old references |
|
2 |
The chapter on milk components is poorly organized. The key parts of milk are: water, proteins (the focus of this review), lactose and fat. Of the protein component, over half is casein in bovine milk. The rest are comprised of various whey proteins. Many methods separate the casein from whey and treat these components separately to yield different peptides. This should be clearly articulated in the article (as shown in figure 2). |
Incorporated as suggested. Kindly refer line no 357-365
|
|
3 |
It would be preferable to have a paragraph or two discussing the potential roles and applications for the infant formula industry, which is a growing sector in many parts of the world. |
Separate section (No. 7) has been added. Kindly refer Line no 974-1005 |
|
4 |
Milk fat globule membranes (MFGM) is another component, with a mix of proteins and lipids, that is of potential use as a value-added product. It is suggested to discuss this in a separate paragraph. |
Separate section (No. 3) has been added. Kindly refer Line no 289-321. |
|
5 |
The section on analytical methods could be condensed and edited for length and clarity. Proper references should take place of excessive details. |
The section has been edited as suggested. Kindly refer line no 559-563, 566-569, 573-576 (deleted) and references have been added. |
|
6 |
It will be preferable to have more details on future directions. Which areas and which food products are most likely to undergo greatest innovations in the next decade? |
Details have been added. Kindly refer Line no. 1019-1021, 1023-1027, 1049-1062. |
|
7 |
It is surprising that milk/dairy comprises 25-30% of diet (as stated in line 141). Please provide further references and discuss what populations these figures are based on. It seems extremely high. |
Rectified. Kindly refer line no 144-145. References has been added (44-46) |
|
8 |
Anti-hypertensive: Please discuss the roles of milk derived tripeptides VPP and IPP which have been commercialized and tested in clinical studies for their BP lowering properties. |
Roles of milk derived tripeptides VPP and IPP have been incorporated. Kindly refer line 208-216. |
|
9 |
Anti-diabetic: Many milk-based peptides (including IPP, VPP) have shown promise as insulin mimetic and potential anti-diabetic agents. These should be described under a separate chapter (instead of brief mention under opioid peptides, since the anti-diabetic effect is not necessarily mediated through opioid receptors). |
Agreed
|
|
10 |
The introduction should refer to several recent reviews on food derived bioactive peptides to provide a better overview to the readers. |
Recent reviews have been cited. Kindly refer line no 79-80 (reference no- 10), 84 (Reference no. 12-14), 95-96 (Reference no. 17-19) |
Reviewer 2 Report
Generally, the manuscript was carefully prepared and has a great potential. My only recommendation is to add a short separate section about the factors that may influence the composition and the amount of bioactive peptides in milk, and the possible differences among the milk products from other ruminant species.
Author Response
Pointwise Answers to the reviewer’s comments
|
S.No |
Comment |
Reply |
|
|
Reviewer #2: |
|
|
1 |
Generally, the manuscript was carefully prepared and has a great potential. My only recommendation is to add a short separate section about the factors that may influence the composition and the amount of bioactive peptides in milk, and the possible differences among the milk products from other ruminant species. |
Separate section (No.4) has been added. Kindly refer line no. 323-354 |